# Elucidating the Immune Response to SARS-CoV-2: Natural Infection versus Covaxin/Covishield Vaccination in a South Indian Population

**DOI:** 10.3390/v16081178

**Published:** 2024-07-23

**Authors:** Agalya Vanamudhu, Renuka Devi Arumugam, Arul Nancy, Nandhini Selvaraj, Kadar Moiden, Syed Hissar, Uma Devi Ranganathan, Ramalingam Bethunaickan, Subash Babu, Nathella Pavan Kumar

**Affiliations:** 1Department of Immunology, ICMR, National Institute for Research in Tuberculosis, Chennai 600031, India; 2National Institutes of Health, National Institute for Research in Tuberculosis, International Center for Excellence in Research, Chennai 600031, India

**Keywords:** Covaxin, Covishield, SARS-CoV-2, COVID-19, T cells, B cells, monocytes and cytokines

## Abstract

A natural infection or a vaccination can initially prime the immune system to form immunological memory. The immunity engendered by vaccination against COVID-19 versus natural infection with SARS-CoV-2 has not been well studied in the Indian population. In this study, we compared the immunity conferred by COVID-19 vaccines to naturally acquired immunity to SARS-CoV-2 in a South Indian population. We examined binding and neutralizing antibody (NAb) levels against the ancestral and variant lineages and assessed the ex vivo cellular parameters of memory T cells, memory B cells, and monocytes and finally measured the circulating cytokine response. COVID-19 vaccination stimulates heightened levels of IgG antibodies against the original strain of SARS-CoV-2, as well as increased binding to the spike protein and neutralizing antibody levels. This enhanced response extends to variant lineages such as B.1.617.2 (Delta, India), B.1.1.529 (Omicron, India), B.1.351 (Beta, South Africa), and B.1.1.7 (Alpha, UK). COVID-19 vaccination differs from SARS-CoV-2 infection by having increased frequencies of classical memory B cells, activated memory B and plasma cells, CD4/CD8 T cells of effector memory, effector cells, stem cell-like memory T cells, and classical and intermediate monocytes and diminished frequencies of CD4/CD8 T cells of central memory and non-classical monocytes in vaccinated individuals in comparison to those with natural infection. Thus, COVID-19 vaccination is characterized by enhanced humoral responses and robust activation of innate and memory T cell responses in comparison to natural infection in a South Indian population.

## 1. Introduction

COVID-19 (coronavirus disease 2019), caused by SARS-CoV-2 infection (severe acute respiratory syndrome coronavirus 2), has been the key reason behind the biggest health crisis all around the world [1]. Nearly three years into the COVID-19 pandemic, multiple COVID-19 vaccines had received approval by governing authorities and the WHO on the basis of vaccine efficacy results from randomized controlled trials. India commenced COVID-19 vaccine administration on 16 January 2021. As of 4 March 2023, over 2.2 billion doses of vaccines, including first, second, and precautionary (booster) doses, had been administered across the country. In India, 95% of the eligible population aged 12 and above have received at least one dose, and 88% are fully vaccinated. Covishield constitutes the majority of the approved vaccines administered in India, promoting our interest in studying its impact, together with Covaxin, on host immune protection. The overall administration of COVID-19 vaccines has markedly lessened the infection rate, severity, and mortality of this disease [2,3]. Although new SARS-CoV-2 variants of concern (VOCs) have diminished the protective immunity produced by natural infection and current vaccines have good neutralizing activity against the variants, the persistence of protective immunity is still unclear [4,5].

Viral immunity is facilitated by immunological memory that is acquired after an initial immune response stimulated by a viral antigen. During natural SARS-CoV-2 infection, prompt and efficient innate and adaptive immune responses develop against the virus to protect the host. Nevertheless, the duration of this SARS-CoV-2-specific dependable protective immunity in individuals with past infection remains poorly understood [6]. COVID-19 vaccines work differently by introducing antigens which have unique features of the SARS-CoV-2 virus to the immune system. The antigen triggers a specific immune response, and this response builds the immune memory so the body can fight off SARS-CoV-2 in the future and continued active immunization can generate herd immunity [6,7] in the population.

In this study, we focused on two of the widely administered vaccines in India. Covaxin/BBV152 is a whole virion inactivated vaccine formulated with a Toll-like receptor ligand adsorbed to alum [8,9], and Covishield (ChAdOx1) is a recombinant, replication-deficient chimpanzee adenovirus vector that encodes SARS-CoV-2 spike glycoprotein [10]. Published reports from clinical trials have demonstrated that both types of vaccines generate high levels of neutralizing antibodies and reduce severe outcomes. Covishield shows an effectiveness of nearly 90%, whereas Covaxin has an effectiveness of about 80%, and both vaccines in India have so far established satisfactory efficacy against numerous mutant variants of SARS-CoV-2 [11]. In this study, we characterized the effectiveness of the Covishield and Covaxin vaccines in comparison to naturally SARS-CoV-2-infected individuals in terms of both innate and adaptive immune responses.

## 2. Materials and Methods

### 2.1. Study Procedure

A prospective cross-sectional study was conducted. The study recruited individuals who received BBV152/Covaxin (manufactured by Bharat Biotech, Hyderabad, in collaboration with the Indian Council of Medical Research, India) and ChAdOx1 nCoV-19/Covishield (manufacturer/developer: AstraZeneca, Serum Institute of India) at vaccination centers in Chennai, India, between November 2022 and May 2023. All adult participants aged more than 18 years and less than 60 years who received two doses of BBV152/Covaxin and ChAdOx1 nCoV-19/Covishield and individuals naturally infected with COVID-19 within 30 days of RT-PCR confirmation who had either asymptomatic, mild, or moderate but not severe disease were enrolled from the Greater Chennai Corporations Urban Health Centers and were eligible to participate in this study. At the time of enrollment, blood samples were collected in sodium heparin and EDTA tubes. Collected blood samples were transported within 2 h to the immunology lab for processing and storage. The demographic profile of the study population is shown in Table 1.

### 2.2. Ex Vivo Immunophenotyping

All antibodies used in the study were from BD Biosciences (San Jose, CA, USA), BD Pharmingen (San Diego, CA, USA), eBioscience (San Diego, CA, USA), or R&D Systems (Minneapolis, MN, USA). Whole blood was used for ex vivo phenotyping, which was performed on n = 76 individuals (vaccinated, n = 51; COVID-19-positive, n = 26). Briefly, in 250 μL aliquots of whole blood, a cocktail of monoclonal antibodies specific for various immune cell types were assessed. Memory T cell phenotyping was performed using antibodies directed against CD45 Peridinin chlorophyll protein (PerCP), CD3 phycoerythrin (PE) Cy7, CD4 allophycocyanin-H7 (APC-H7), CD8 AmCyan, CD28 APC, CD45RA Pacific Blue, CCR7-FITC, and CD95 PE. Naive cells were classified as CD45RA+ CCR7+ CD95- CD28+, central memory (CM) cells as CD45RA- CCR7+ CD95+ CD28+, effector memory (EM) cells as CD45RA-CCR7- CD95+ CD28, terminal effector (TEM) cells as CD45RA- CCR7- CD95+ CD28-, and stem cell memory (SCM) as CD45RA+ CCR7+ CD95+ CD28+ [12]. Memory B cells were classified as follows: Naive B as CD45+ CD19+ CD21+ CD27, classical memory B cells as CD45+ CD19+ CD21+ CD27+, activated memory B cells as CD45+ CD19+ CD21- CD27+, atypical memory B cells as CD45+ CD19+ CD21-CD27-, immature B cells as CD45+ CD19+ CD21+ CD10+, and plasma cells as CD45+ CD19+ CD21- CD20 [13]. Monocytes were classified as follows: classical monocytes as CD45+ HLA-DR+ CD14^hi^ CD16-, intermediate monocytes as CD45+ HLA-DR+ CD14^hi^ CD16^dim^, and non-classical monocytes were classified as CD45+ HLA- DR + CD14^dim^ CD16^hi^. Eight-color flow cytometry was performed on a FACS Canto II flow cytometer with FACS DIVA software, version 6 (Becton Dickinson, San Diego, CA, USA). The gating was set by forward and side scatter, and 100,000 gated events were acquired. Data were collected and analyzed using FLOW JO software 10.8.0 (TreeStar, Ashland, OR, USA). Leukocytes were gated using CD45 expression versus side scatter [14,15].

### 2.3. Multiplex ELISA

Circulating levels of IFNγ, IL-2, TNFa, IL-17A, IL-6, IL-12, IL-1α, IL-1β, IL-4, IL-5, IL-10, IFNα, and IFNβ were measured using a Luminex Human Magnetic multiplex assay kit (R&D Systems). The lowest detection limits were as follows: IFNγ, 6.49 pg/mL; IL-2, 3.55 pg/mL; TNFa, 9.12 pg/mL; IL-17A, 8.93 pg/mL; IL-6, 8.92 pg/mL; IL-12, 19.45 pg/mL; IL-1α, 10.46 pg/mL; IL-1β, 3.75 pg/mL; IL-4, 1.12 pg/mL; IL-5, 6.3 pg/mL; IL-10, 34.9 pg/mL; IFNα, 4.39 pg/mL; IFNβ, 3.61 pg/mL. The lowest standard value was assigned to the samples that were below the threshold of detection. Forty-four samples were analyzed in duplicate, and lab personnel were blinded to clinical groups. All the assays were performed according to the manufacturer’s instructions.

### 2.4. Antibody Assays

Serological testing for targeting the viral spike protein IgG (S) was performed using a YHLO iFlash 1800 Chemiluminescence Immunoassay Analyzer using iFlash-SARS-CoV-2 IgG (S). The cut-off value for SARS-CoV-2 IgG (according to the manufacturer) was >10 AU/mL; IgG concentrations >10.00 AU/mL were considered positive and those <10.00 AU/mL were considered non-reactive. Plasma samples were used to measure the circulating neutralizing antibody levels using the SARS-CoV-2 Surrogate Virus Neutralization Test Kit (sVNT) according to the manufacturer’s (GenScript) instructions. The sVNT Kit detects neutralizing antibodies against SARS-CoV-2’s spike protein by competitively binding with a labeled spike protein to the receptor-binding domain (RBD). It assesses the ability of antibodies to block viral entry, providing a rapid and safe alternative to traditional virus neutralization assays. The sVNT detects NAbs without the need to use live virus or cells and can be completed in 1–2 h in a BSL2 lab. A value of more than or equal to 20% inhibition was considered positive and <20% was considered non-reactive for SARS-CoV-2 NAb detection. (The term “inhibition” refers to the ability of antibodies present in a sample to block the interaction between the receptor-binding domain (RBD) of the spike protein of SARS-CoV-2 and its receptor on the host cell.)

## 3. Statistical Analysis

Geometric means were used for measurements of central tendency. Statistically significant differences between the vaccinated and COVID-19-infected groups were analyzed by nonparametric Mann–Whitney U test with Dunn’s multiple comparisons. Analyses were performed using GraphPad PRISM Version 9.0 (GraphPad Software, La Jolla, CA, USA).

## 4. Results

### 4.1. COVID-19 Vaccinated and Infected Cohort

To compare the host immune responses and immune memory, we enrolled subjects who were vaccinated against COVID-19. All enrolled participants received homologous vaccines: one group received Covaxin, while the other received Covishield, completing their two-dose vaccination schedules (the Indian government has recommended that the time interval between the 1st and 2nd doses for Covaxin should be 4 weeks and that for Covishield should be between 12 and 16 weeks) with no previous RT-PCR or rapid antigen test confirmation for the last six months. Individuals naturally infected with COVID-19 were those with a post-PCR confirmation within 30 days who were asymptomatic or had mild or moderate, but not severe, disease when enrolled (according to the guidelines of the Government of India, Ministry of Health and Family Welfare; CLINICAL MANAGEMENT PROTOCOL: COVID-19; Dated: 13 June 2020) and were not vaccinated with either COVID-19 vaccine (Figure 1). In both study groups, complete blood count, IgG spike binding antibodies, neutralizing antibodies for the variants of concern, whole blood immunophenotyping (monocytes, memory T cells, and memory B cells), and circulating pro-inflammatory cytokine levels were measured. The demographics of the vaccinated and naturally COVID-19-infected individuals are shown in Table 1. In addition, a table discriminating the results with vaccine types has been produced (Appendix A). Furthermore, we obtained a cohort of n = 20 historic controls (pre-pandemic healthy individuals), matched for age and sex, to compare immunological responses between vaccinated individuals and those who have had COVID-19 infection.

### 4.2. Alterations of Hematological Parameters in COVID-19 Vaccinated versus Naturally Infected Individuals

To determine the association of hematological parameters following two doses of COVID-19 vaccines (Covaxin and Covishield) in comparison to natural infection, we measured the baseline hematological parameters such as hemoglobin, RBCs, hematocrit, platelets, absolute count of WBCs, neutrophils, lymphocytes, and monocytes (Table 2). Our results showed that the platelet absolute numbers (*p* = 0.0256), neutrophil absolute numbers (*p* = 0.0048), neutrophil percentage (*p* = 0.0070), basophil absolute numbers (*p* = 0.0023), and basophil percentage (*p* = 0.0434) were significantly higher in vaccinated individuals when compared to individuals with natural infection. However, the lymphocyte percentage (*p* = 0.0359) and monocyte percentage (*p* = 0.0078) were significantly lower in vaccinated individuals compared to those with natural infection.

### 4.3. COVID-19 Vaccination Induced Enhanced IgG and Neutralizing Antibodies against SARS-CoV-2 Viral Variants

To examine the nature of humoral immunity following two doses of COVID-19 vaccines (Covaxin and Covishield) in comparison to natural infection, we measured the levels of IgG against the spike (S) antigen. As shown in Figure 2, vaccinated individuals showed enhanced IgG (S) in comparison to naturally infected individuals. Furthermore, the neutralizing antibody levels against the wild-type strain (ancestral strain [Wuhan]) and the variant lineages B.1.617.2 (Delta), B.1.1.529 (Omicron), B.1.351 (Beta, SA), and B.1.1.7 (Alpha, UK) were significantly elevated in the vaccinated group in comparison to naturally infected individuals.

### 4.4. Elevated B Cell Subsets in COVID-19 Vaccinated Individuals

To elucidate the total B cell subset phenotypes following COVID-19 vaccination or infection, we utilized the B cell subsets gating strategy shown in Appendix A. As shown in Figure 3, the frequencies of classical memory (CD45+ CD19+ CD21+ CD27+) and activated memory B cells (CD45+ CD19+ CD21- CD27+) and plasma (CD45+ CD19+ CD21- CD20-) cells were significantly elevated, with a twofold increase among the vaccinated population in comparison to the naturally infection group. However, among the other memory B cell subsets such as immature (CD45+ CD19+ CD21+ CD10+), naïve (CD45+ CD19+ CD21+ CD27-), and atypical memory B cells (CD45+ CD19+ CD21- CD27-), there were no observed differences between the study groups.

### 4.5. CD4+ and CD8+ Memory T Cell Subsets Are Altered by COVID-19 Vaccination

To assess the ex vivo phenotypes of CD4+ memory T cell subsets following COVID-19 vaccination and natural infection, we used the CD4+ T cell subsets gating strategy represented by Appendix A (CD45RA, CCR7, CD95, and CD28 are markers found on T cells: CD45RA marks naive T cells that have not encountered specific threats yet; CCR7 guides T cells to lymph nodes, where they can respond to threats; CD95 controls T cell survival and prevents overactive immune responses; and CD28 helps activate T cells by interacting with other immune cells). As shown in Figure 4, the frequencies of CD4+ effector memory (CD45RA- CCR7- CD95+ CD28), effector (CD45RA- CCR7- CD95+ CD28-), and stem cell-like memory (CD45RA+ CCR7+ CD95+ CD28+) T cell subsets were significantly enhanced, with a twofold increase, while the frequency of central memory cells (CD45RA- CCR7+ CD95+ CD28+) was significantly diminished with a twofold decrease in the vaccinated population in comparison to infected individuals. Similarly, the frequencies of CD8+ effector memory (CD45RA- CCR7- CD95+ CD28), effector (CD45RA- CCR7- CD95+ CD28-), and stem cell-like memory (CD45RA+ CCR7+ CD95+ CD28+) T cell subsets were significantly enhanced, with a twofold increase, while the frequency of the central memory cell (CD45RA- CCR7+ CD95+ CD28+) subset was significantly diminished with a twofold decrease in the vaccinated population in comparison to infected individuals.

### 4.6. Monocyte Subsets Are Altered by COVID-19 Vaccination

We elucidated the ex vivo phenotypes of monocyte subsets following COVID-19 vaccination and natural infection, and a representative flow cytometry plot with the gating strategy for monocyte subsets is shown in Appendix A As shown in Figure 5, within the monocyte subsets, vaccinated individuals exhibited increased frequencies of classical monocytes (CD45+ HLA-DR+ CD14^hi^ CD16-) and intermediate monocytes (CD45+ HLA-DR+ CD14^hi^ CD16^dim^) in comparison to naturally infected individuals. In contrast, the frequencies of non-classical monocytes (CD45+ HLA- DR+ CD14dim CD16hi) were significantly decreased in vaccinated individuals in comparison to naturally infected individuals.

### 4.7. COVID-19 Vaccine Induces Diminished Plasma Levels of Type 1, Type 2, Type 17, and Other Pro-Inflammatory Cytokines

We next examined the plasma levels of type 1, type 2, type 17, and other pro-inflammatory cytokines following COVID-19 vaccination in comparison to natural infection. As shown in Figure 6, the circulating levels of type 1 cytokines such as IFNγ (*p* < 0.0001) and IL-2 (*p* < 0.0001), type 2 cytokines such as IL-4 (*p* = 0.0003) and IL-5 (*p* < 0.0001), the type 17 cytokine IL-17A (*p* = 0.0104), and other pro-inflammatory cytokines (IL-6 [*p* = 0.0005], IL-12 [*p* < 0.0001], IL-1α [*p* = 0.0067], IL-1β [*p* = 0.0471], IFNα [*p* = 0.0039], and IFNβ [*p* = 0.0029]) were significantly diminished in the vaccinated group in comparison to the naturally infected group.

A separate sub-analysis was conducted to compare pre-pandemic healthy controls with individuals vaccinated against SARS-CoV-2 and those who had COVID-19 infection. Our findings clearly showed minimal expression of immune subsets and inflammatory cytokine levels consistently below the threshold limits, suggesting that all observed responses are attributable to SARS-CoV-2 (Appendix A).

## 5. Discussion

Infection with SARS-CoV-2 is characterized by a broad spectrum of clinical syndromes, which range from asymptomatic disease or mild influenza-like symptoms to severe pneumonia and acute respiratory distress syndrome [16]. Understanding the key features and evolution of immune responses to SARS-CoV-2 is essential in forecasting COVID-19 outcomes and for developing effective strategies to control the pandemic. Ascertaining long-term immunological memory against SARS-CoV-2 is also critical to understanding durable protection [16]. The immunopathogenesis of COVID-19 is complex and may be associated with the virulence of SARS-CoV-2 and the lack of temporal coordination between the innate and adaptive immune responses [17]. The rapid development of multiple COVID-19 vaccines has been a triumph of biomedical research, and billions of vaccine doses have been administered worldwide. Inactivated vaccines are considered safe because they do not contain live virus, thus eliminating the risk of causing disease in vaccinated individuals, and also offer safety, stability, and suitability for vulnerable populations but may require multiple doses and adjuvants for optimal efficacy. These inactivated vaccines stimulate immune responses primarily through the production of antibodies by B cells. These responses are crucial for providing protection against specific pathogens without causing the disease itself [18,19]. Recombinant adenovirus vector vaccines can induce potent immune responses because adenoviruses are highly immunogenic. They can stimulate both cellular (T-cell-mediated) and humoral (antibody-mediated) immune responses, providing comprehensive protection against pathogens. Adenovirus vectors can be engineered to carry genes encoding specific antigens from various pathogens; however, in people with pre-existing immunity to adenoviruses due to prior natural infections or vaccinations, this immunity can potentially reduce the effectiveness of adenovirus vector vaccines by neutralizing the vector before it can deliver the vaccine antigen [20,21]. In India, the roll-out of Covaxin and Covishield was initiated among all adult age groups. For all vaccine candidates and naturally infected individuals, it is important to understand the functional immune response for robust and lasting protection and the durability of responses involving neutralizing antibodies. In addition, there is a gap in the understanding of the immune correlates of protection and the long-term immune memory from natural infection and vaccination by VOCs in COVID-19 vaccinated vs. naturally infected individuals. A recent study also reported that, overall, while natural infection can provide immunity against SARS-CoV-2, COVID-19 vaccination offers a more reliable and controlled means of achieving robust and durable protection against the virus and its variants [22]. In this study, we elucidated the immunological responses towards COVID-19 vaccination and infection.

The correlates of protective immunity to SARS-CoV-2 are not completely understood, but in terms of humoral immunity based on B cell responses, antibody-mediated responses including binding and neutralizing antibody (NAb) effects are very important [23]. We previously reported that Covaxin induces enhanced SARS-CoV-2-specific binding antibodies of IgG against both the spike (S) and nucleocapsid (N) and elevated NAb levels against the ancestral strain (Wuhan, wild-type) and other VOCs, such as Delta, Delta Plus, Beta, Alpha, at M1, M2, M3, M4, M6, and M12 in comparison to M0, indicating an enhanced humoral immune response with persistence until at least 12 months post-vaccination [24]. In this current study, we have evaluated the potential of binding and neutralizing antibodies generated in individuals who received COVID-19 vaccines (Covishield and Covaxin) and unvaccinated, naturally infected individuals. Our data revealed that levels of SARS-CoV-2 binding and neutralizing Abs against the wild-type virus and VOCs were clearly elevated in vaccinated individuals in comparison to naturally infected individuals, indicating that the humoral response generated by the vaccines is effective in generating immunity [20,25]. We speculate that the increased antibody level in the vaccinated group was probably due to the additive effect of the adjuvant used in the vaccines, which may have induced a specific immune response that may be robust and long-lasting; on the other hand, all the enrolled naturally infected individuals had either mild to moderate infection, which in turn led to the low circulating neutralizing antibody titers and delay in mounting memory responses.

While antibodies are the main correlates of protection against infection and vaccination [26], T and B cell memory responses are equally important for protective immunity [27], and cellular immunity might also play a vital role in modulating the disease severity and resolving the infection [28]. The role of B cells in viral infection is dynamic with the function of cytokine production, antigen presentation, and antibody secretion [29]. The memory B cell response to COVID-19 progresses for 1–6 months after infection in a manner that is consistent with antigen persistence [7,30,31]. In this study, we found that classical memory B cells, activated memory B cells, and plasma cells were significantly elevated in the vaccinated group in comparison to infected individuals, revealing that the B cell memory antibody response was higher in the vaccinated population and suggesting that the vaccines may have been beneficial in protecting against COVID-19. In agreement with our study, it has been reported that existing memory B cells all are elevated after vaccination, and the degree of enhancement is correlated with the number of memory B cells pre-existing in the SARS-CoV-2-infected individuals [32,33].

T cell responses of the adaptive immune system pick up during an initial immune response and have an impact on the development of immunological memory [34]. An increased number of antigen-specific memory cells that quickly express effector molecules following antigen re-exposure characterizes protective immunity produced in response to infection or vaccination. Memory cells are functionally important for the priming event to prevent or control reinfection [35]. The goal of the current study was to compare the strength of vaccination-induced immunological T cell memory with two distinct vaccine delivery systems in comparison to that following infection. To improve our comprehension of the protection provided by the various COVID-19 vaccines and infection and comprehend the fundamental variations in immunogenicity and immune memory, a direct, side-by-side, comprehensive evaluation of effector and memory immune responses induced by various vaccine platforms was essential.

In our current study, we focused on using CD45RA and CCR7 to primarily delineate memory T cell differentiation states, rather than relying on CD44 and CD62L, which predominantly indicate T cell activation and tissue homing capabilities. The rationale behind this choice lies in the critical role of CD45RA and CCR7 as markers for defining distinct subsets of memory T cells. These markers provide essential insights into the differentiation, functional capabilities, and immune response roles of memory T cells. Furthermore, their utilization aids in deepening our fundamental understanding of adaptive immune responses in both health and disease contexts. Published data indicate that COVID-19 vaccines can elicit robust CD4 T cell memory responses, and the mRNA-1273 vaccine generated spike-specific memory CD4 T cell frequencies that were greater in vaccinated individuals than in previously infected individuals [35]. Nearly 100% of people showed spike-specific CD4 T cell responses weeks after receiving two doses of the mRNA COVID-19 vaccine, and nearly 100% of people still showed memory CD4 T cell responses six months after the second dose [35,36]. Consistent with previous published studies, our study also indicated that the percentage frequencies of CD4+ effector memory, effector, and stem cell-like memory T cells were increased, whereas the frequency of central memory cells was decreased in the vaccinated group in comparison to the infected group.

In general, CD8+ T lymphocytes play a crucial role in the management and eradication of viral infections. Multiple lines of evidence, in particular, suggest CD8+ T cells as part of the overall adaptive immune response to SARS-CoV-2 [37]. Findings from studies on CD8+ memory T cell responses to vaccination showed that approximately 70%–90% of people exhibited spike-specific CD8+ T cell responses weeks after receiving two doses of an mRNA vaccination for COVID-19, and memory CD8+ T cells were found in 41%-65% of people six months following the second dosage (7 months from first dose) [35,36]. Multiple other findings also reported lower SARS-CoV-2-specific CD8+ T cell responses in individuals hospitalized with COVID-19, which is consistent with a weak CD8+ T cell response, predisposing people to more severe COVID-19, particularly in older adults with fewer naive CD8 T cells [38,39]. In agreement with these reports, our findings revealed that the percentage frequencies of CD8+ effector memory, effector, and stem cell-like memory T cells were increased, whereas the frequency of central memory cells was decreased in the vaccinated group in comparison to the infected group, indicating that the vaccinated individuals acquired stronger memory T cell immunity as protection against the symptomatic disease.

Significant contributions to the hyperinflammation seen in COVID-19-affected individuals with severe disease have been attributed to monocytes and macrophages. The production of cytokines and inflammatory markers by monocytes, which can recognize, process, and deliver antigens to T cells, allows them to control the immune response [40]. However, their contribution to the successful control of SARS-CoV-2 through vaccination and infection remains poorly understood. Recent studies have reported that COVID-19 patients with severe disease showed greater monocyte counts, higher proportions of classical monocytes, and lower proportions of intermediate and non-classical monocytes in comparison to patients with mild disease [41]. In addition, it was also reported that acute and convalescent COVID-19 patients displayed dynamic changes in monocyte subset frequencies and activation status [42]. Our findings revealed that vaccinated individuals showed elevated frequencies of classical and intermediate monocytes and diminished frequencies of non-classical monocytes in comparison to individuals with COVID-19 infection, indicating that the altered frequencies of monocytes are characteristic features of COVID-19.

The pathophysiology of COVID-19 involves inflammation, and when the negative effects of the immune response to infection outweigh the immediate antiviral benefit, a ‘cytokine storm’ during the acute phase of SARS-CoV-2 infection may be fatal [43]. Indeed, it has been reported that the increase in cytokine levels during the acute phase is linked to a higher risk of illness severity and mortality [44]. Responses to vaccination and infection largely rely on cytokine responses. The relationship between SARS-CoV-2 immunization and cytokine levels and trajectories in people with COVID-19 infection is not well understood, either in the short or long term [45]. This study evaluated the longitudinal association of vaccination against SARS-CoV-2 on cytokine levels among adults with natural SARS-CoV-2 infection. Prior studies have reported that cytokines are associated with disease severity related to SARS-CoV-2 infection [44]. On the other hand, vaccination is efficient in reducing morbidity and mortality from COVID-19 [2]. Though the effect of vaccination on the changes in levels of cytokines over time among infected individuals is not clear, our study found that recently vaccinated individuals (two doses of the vaccine) had lower levels of most of the cytokines than those who were infected with COVID-19 (unvaccinated). These findings imply that immunization may protect against inflammation even in the presence of breakthrough symptoms of infection. However, our study has certain limitations: the sample size was moderate, the study was performed in a single cohort without external validation, and cause–effect relationships were not determined.

In conclusion, our findings revealed that both vaccines (Covishield and Covaxin) are more effective toward the wild-type virus and VOCs. This analysis confirmed that vaccine-mediated immune memory responses afford long lasting and stronger protection compared to natural COVID-19 infection in unvaccinated individuals. In the future, further studies are warranted to elucidate the antigen-specific T cell-mediated pathways that may give broad-spectrum protection, which would help us to understand the humoral and cell-mediated immunity, together with exploring the immunogenetics component, which can have a greater influence in preempting the disease and evolving symptoms. In summary, both vaccination and natural infection confer some degree of immunity against COVID-19. Vaccination may provide a more controlled and predictable immune response, whereas natural infection can result in more variable immune outcomes. Following up with these individuals is crucial to understanding the long-term effectiveness and durability of immunity in both vaccinated and naturally infected individuals, especially in the context of emerging variants of the virus.

## Figures and Tables

**Figure 1 viruses-16-01178-f001:**
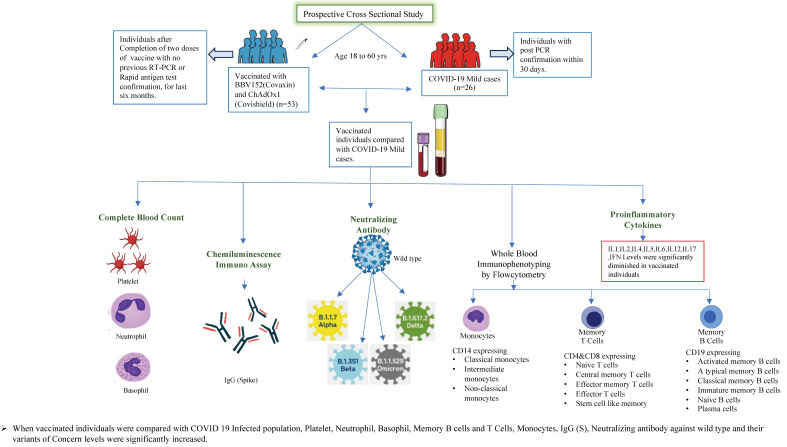
COVID-19 vaccinated and infected cohorts. Study cohort characteristics. The timeline of immunizations and immune assays performed.

**Figure 2 viruses-16-01178-f002:**
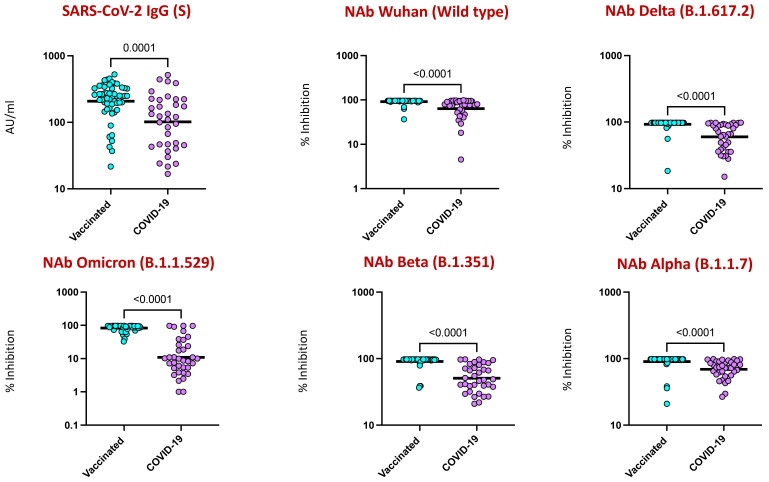
SARS-CoV-2 binding and neutralizing antibody response to COVID-19 vaccines and in individuals with COVID-19 infection. The plasma levels of SARS-CoV-2 binding antibodies of IgG [S] and Surrogate Virus Neutralization for wild-type and variant lineages of B.1.617.2 (Delta), B.1.1.529 (Omicron), B.1.351 (Beta, SA), and B.1.1.7 (Alpha, UK) in those with COVID-19 vaccination and COVID-19 infection. The data are represented as scatter plots, with each circle representing a single individual. The *p*-values were calculated using the non-parametric Mann–Whitney U test and with Holms correction for multiple comparisons.

**Figure 3 viruses-16-01178-f003:**
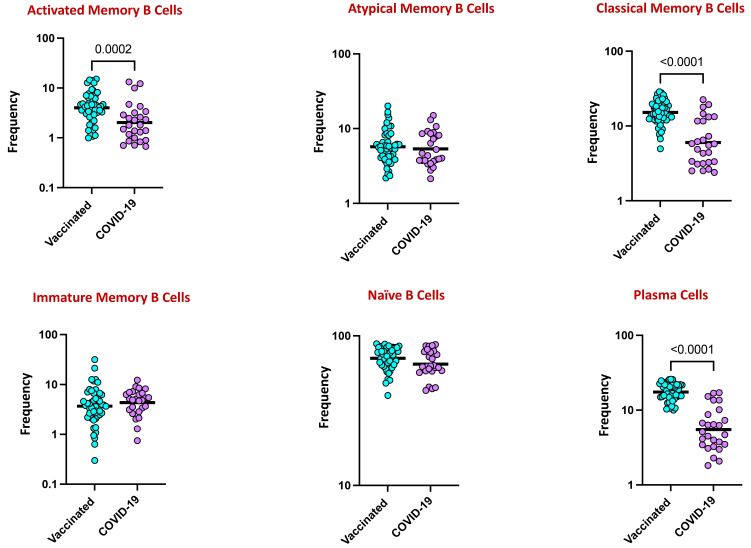
Elevated B cell subsets in COVID-19 vaccinated individuals. Frequencies of B cell subsets in those with COVID-19 vaccination and COVID-19 infection. The data are represented as scatter plots, with each circle representing a single individual. The *p*-values were calculated using the non-parametric Mann–Whitney U test and with Holms correction for multiple comparisons.

**Figure 4 viruses-16-01178-f004:**
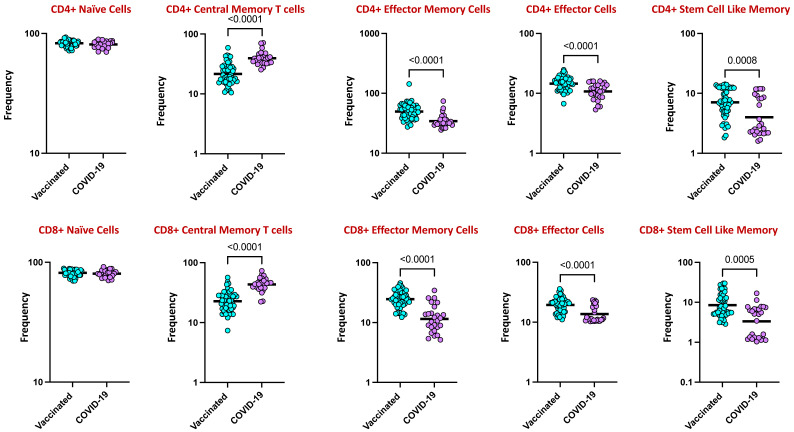
Frequencies of T cell subsets in COVID-19 vaccinated individuals and in individuals with COVID-19 infection. The data are represented as scatter plots, with each circle representing a single individual. The *p*-values were calculated using the non-parametric Mann–Whitney U test and with Holms correction for multiple comparisons.

**Figure 5 viruses-16-01178-f005:**
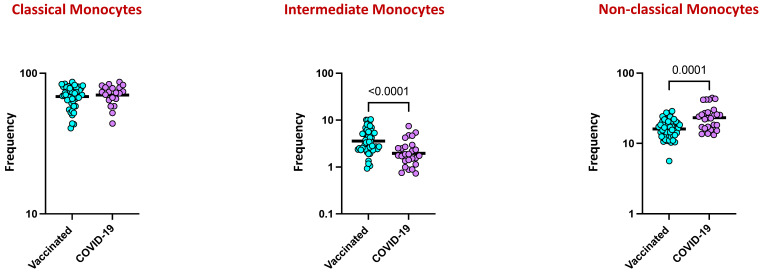
Frequencies of monocyte subsets in COVID-19 vaccinated individuals and individuals with COVID-19 infection. The data are represented as scatter plots, with each circle representing a single individual. The *p*-values were calculated using the non-parametric Mann–Whitney U test and with Holms correction for multiple comparisons.

**Figure 6 viruses-16-01178-f006:**
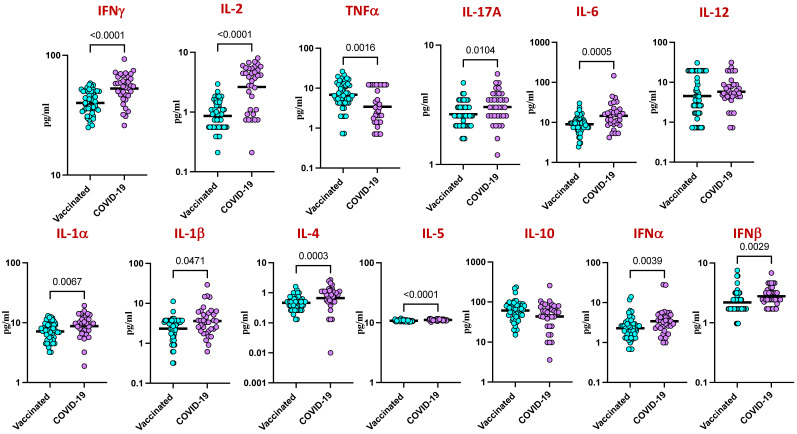
Diminished inflammatory cytokine responses in COVID-19 vaccinated individuals. Plasma levels of a panel of inflammatory cytokines were measured in individuals with COVID-19 vaccination or COVID-19 infection. The data are represented as scatter plots, with each circle representing a single individual. The *p*-values were calculated using the non-parametric Mann–Whitney U test and with Holms correction for multiple comparisons.

**Table 1 viruses-16-01178-t001:** Demographic profile of study population.

S. No	Characteristics of Study Population	Vaccinated (n = 53)	COVID-19-Positive Cases (n = 26)
1	Age	41.32 (18–59)	35 (21–58)
2	Male	25	15
3	Female	28	11
4	COVID-19	Negative	Positive
5	Tuberculosis	Negative	Negative
6	Diabetes mellitus	15/38	9/17
7	Hypertension	9/44	5/21
8	Cardiovascular disease	5/48	Nil
9	Chronic obstructive pulmonary disease (COPD)	Nil	Nil
10	Dyslipidemia	Nil	Nil
11	Immunological disease	Nil	Nil
12	Autoimmune disease	Nil	Nil
13	SARS-CoV-2 IgG (S) (AU/ML)	255.58 (21.68–526.62)	115.285 (16.82–514.97)
14	NAb_Wild type (% of inhibition)	92.54 (36.38–97.75)	76.305 (4.52–96.77)
15	NAb_Delta (% of inhibition)	94.19 (18.47–97.88)	65.755 (28.18–97.93)
16	NAb_Omicron (% of inhibition)	81.47 (0.2–97.39)	8.12 (−17.75–97.44)
17	NAb_SA (% of inhibition)	89.64 (13.3–97.7)	48.375 (20.9–97.6)
18	NAb_UK (% of inhibition)	93.13 (20.95–98.27)	72.05 (29.55–98.19)

**Table 2 viruses-16-01178-t002:** Hematological parameters.

S. No	Hematology	Vaccinated (n = 53)	COVID-19 (n = 26)	*p*-Value
1	WBC 10^3^/mL	8.52 (4.47–14.13)	7.65 (2.12–10.5)	0.1153
2	RBC 10^6^/mL	4.85 (3.13–6.11)	4.93 (3.71–7.26)	0.7126
3	HGB g/dL	13.61 (9.42–18.06)	14.165 (9.95–19.9)	0.4695
4	MCV fL	84.94 (68.4–104.1)	85.6 (59–94.4)	0.4764
5	MCH Pg	28.06 (20.6–37.7)	28.6 (18.7–32.6)	0.7937
6	MCHC g/dL	32.99 (30.1–39.5)	33.3 (31.1–34.6)	0.3167
7	HCT%	41.19 (29.9–53.3)	42.2 (31.6–59)	0.4905
8	PLT 10^3^/mL	287.24 (127.9–455.1)	241.4 (129.8–350.9)	**0.0256**
9	Neutrophil 10^3^/mL	4.60 (2.08–10.68)	3.035 (0.76–5.8)	**0.0048**
10	Lymphocyte 10^3^/mL	3.15 (1.77–4.98)	3.095 (1.17–5.75)	0.6917
11	Monocyte 10^3^/mL	0.47 (0.24–0.88)	0.53 (0.02–2.16)	0.0633
12	Eosinophil 10^3^/mL	0.21 (0.04–0.71)	0.15 (0.05–0.55)	0.5524
13	Basophil 10^3^/mL	0.07 (0.03–0.15)	0.05 (0–0.1)	**0.0023**
14	Neutrophil%	52.90 (35.01–76.48)	38.245 (20.7–65.47)	**0.0070**
15	Lymphocyte%	37.87 (19.94–54.14)	42.245 (25.99–64.6)	**0.0359**
16	Monocyte%	5.76 (2.7–8.95)	6.82 (1.15–28)	**0.0078**
17	Eosinophil%	2.53 (0.33–7.37)	1.8 (0.7–14.06)	0.8682
18	Basophil%	0.93 (0.36–1.84)	0.645 (0–1.8)	**0.0434**

Statitically significant differences between the study groups, if the *p*-Value is less than 0.005, are shown in bold.

## Data Availability

All the reported data are available within the manuscript.

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
