# Peer review of "Elucidating the Immune Response to SARS-CoV-2: Natural Infection versus Covaxin/Covishield Vaccination in a South Indian Population"

_viruses, 2024, doi:10.3390/v16081178_

Round 1

Reviewer 1 Report

Comments and Suggestions for Authors

The manuscript by Agalya and colleagues evaluates the immune responses in two populations of people that: a) were vaccinated against SARS-CoV-2 (53 people); and b) had natural SARS-CoV-2 infections (26 people). Vaccinated individuals were given two doses of ChAdOx1 nCoV-19/Covishield (a replication-deficient chimpanzee adenovirus vector expressing the Spike protein) and BBV152/Covaxin (an inactivated whole virus vaccine). There are several concerns for the studies presented here including the use of a vaccine that has been taken off the market (I assume for potential safety reasons), the lack of details on the vaccinations, and the lack of control populations. My comments are listed below.

Major comments:

Comment 1. The details of the vaccination should be clearer. For example, were the two vaccines given together or separately? Were individuals either given the Covaxin or CoviShield vaccine? What was the time interval between the first and second doses? This is important as in lines 130-131 the authors discuss patients receiving “two doses of Covaxin or Covishield vaccine.”

Comment 2. The ChAdOx1 nCoV-19/Covishield has been withdrawn based on concerns of thrombotic thrombocytopenia. Thus, a relevant question is, “Why are the authors analyzing immune responses from a vaccine withdrawn from the market?”

Comment 3. The authors analyze virus-neutralization responses using a SARS-CoV2 Surrogate Virus Neutralization test kit. In Figure 2, the data is presented as % inhibition. The authors need to define the meaning of 100% inhibition. How does this compare to titers obtained with virus neutralization test titers?

Comment 4. In Table 1, the values stated for ‘S.No” rows 15-20 are what kind of units? Please define in the Materials and methods.

Comment 5. In Figures 3-5, the authors quantify the levels of different subsets of B cells (Fig.3), T cells (Fig. 4) and monocytes (Fig. 5). On the y-axis is the frequency of these cells. For example, the scale may be from 1 to 100. What is missing the frequency of what? Is it 10 or 100 cells per 1,000,000? Without this information, the data has no meaning. This needs to be fixed. Additionally, the experiments in these figures are missing important controls of non-vaccinated, non-infected individuals and individuals vaccinated with an unrelated virus. Finally, while the authors can find some differences, these differences may be due to natural variations in the human population. Functional anti-SARS-CoV-2 assays in various cell populations would be helpful to correlate with these differences.

Comment 6. In Figure 6, the authors show cytokine levels in vaccinated versus natural infections. The authors show elevated levels of most cytokines in the COVID-19 group. While these authors have a p-value, most levels are minimal at best and are likely not biologically significant. Similar to Figures 3-5, the experiments reported in Figure 6 are missing important controls of non-vaccinated, non-infected individuals.

Minor comments:

1. Lines 30-32: “COVID-19 (Corona Virus Disease-2019) caused by infection of SARS-CoV-2 (Severe Acute Respiratory Syndrome causing Corona Virus – 2) has been the key reason behind 31 the utmost health crisis all around the world [1].”

should be changed to,

“COVID-19 (Corona Virus Disease-2019) caused by infection of SARS-CoV-2 (Severe Acute Respiratory Syndrome Coronavirus 2) has been the key reason behind 31 the utmost health crisis all around the world [1].”

2. Lines 41-43: The sentence, “During the natural SARS-CoV-2 infection, immunity which develops results in prompt and efficient immune response, thereby protecting the host.”

should be rewritten as,

During the natural SARS-CoV-2 infection, prompt and efficient innate and adaptive immune responses develop against the virus to protect the host.”

3. Line 72: “enrolment” is spelled “enrollment.”

4. Line 74:  As used, “Immunology lab” should not be capitalized.

5. Line 98: Should “1,00000” be “100,000”?

6. Lines 108-109: The sentence “N = 44 samples were analyzed per batch in duplicates, and lab personal were blinded to the clinical groups.”

should be re-written as,

Forty-four samples were analyzed in duplicate and lab personnel were blinded to clinical groups.

7. Lines 114-116: The sentences, “The cut-off value for SARS-CoV-2 IgG, according to the manufacturer, IgG concentrations more than or equal to 10.00 AU/mL was considered as positive and <10.00 115 AU/mL was considered as non-reactive,” 

should be rewritten as,

 “The cut-off value for SARS-CoV-2 IgG (according to the manufacturer) was >10 AU/mL, IgG concentrations >10.00 AU/mL was considered as positive and  <10.00 AU/mL considered as non-reactive.”

8. Lines 130-134: The sentence, “COVID-19 vaccinated individuals: Individuals after Completion of two doses of Covaxin or Covishield vaccine with no previous RT-PCR or Rapid antigen test confirmation, for last six months and COVID-19 naturally infected individuals: Individuals with post PCR confirmation within 30 days, who had either asymptomatic, mild and moderate but not, severe disease individually are enrolled…”

should be rewritten as,

 COVID-19 vaccinated individuals were those who completed two doses of Covaxin or Covishield vaccine with no previous RT-PCR or Rapid antigen test confirmation, for the last six months and COVID-19 naturally infected individuals were those with a post-PCR confirmation within 30 days, who were asymptomatic or had mild and moderate, but not, severe disease individually when enrolled…..”

The authors discuss “COVID naturally infected” individuals as having asymptomatic, mild, or moderate infections of SARS-CoV-2 yet in Figure 1, the flowchart only mentions individuals having mild infections. Is this correct? If so, this should be stated in the Materials and Methods section. Why didn’t the authors analyze individuals with asymptomatic and moderate infections? These also would be important to examine.

9. Lines 175,  211, 242: “p value….” should be changed to “The p-value……”

Lines 251-253:  The sentence, “The immunological pathogenicity of COVID-19 is difficult and may be correlated with the virulence of SARS-CoV-2 and the lack of the temporal coordination between innate and adaptive immune responses [17].”

would sound better as,

The immunopathogenesis of COVID-19 is complex and may be associated with the virulence of SARS-CoV-2 and the lack of temporal coordination between the innate and adaptive immune responses [17].

10. Line 255:  “Covid-19” should be “COVID-19.”

Comments on the Quality of English Language

The quality of the English needs some work. 

Author Response

Reviewer 1 :

The manuscript by Agalya and colleagues evaluates the immune responses in two populations of people that: a) were vaccinated against SARS-CoV-2 (53 people); and b) had natural SARS-CoV-2 infections (26 people). Vaccinated individuals were given two doses of ChAdOx1 nCoV-19/Covishield (a replication-deficient chimpanzee adenovirus vector expressing the Spike protein) and BBV152/Covaxin (an inactivated whole virus vaccine). There are several concerns for the studies presented here including the use of a vaccine that has been taken off the market (I assume for potential safety reasons), the lack of details on the vaccinations, and the lack of control populations. My comments are listed below.

Major comments:

Comment 1. The details of the vaccination should be clearer. For example, were the two vaccines given together or separately? Were individuals either given the Covaxin or CoviShield vaccine? What was the time interval between the first and second doses? This is important as in lines 130-131 the authors discuss patients receiving “two doses of Covaxin or Covishield vaccine.”

Reply : We thank reviewer for bringing this valuable point, as suggested we now added much more clear information about the vaccination status. Changes are made in lines 162-169

We have now modified as “All enrolled participants received homologous vaccines. One group received Covaxin, while another received Covishield, completing their two-dose vaccination schedules.(Indian government has recommended that the time interval between the 1st and 2nd dose for Covaxin should be between 4 weeks and Covishield should be between 12-16 weeks) with no previous RT-PCR or Rapid antigen test confirmation, for the last six months and  COVID-19 naturally infected individuals were those with a post-PCR confirmation within 30 days, who were asymptomatic or had mild and moderate, but not, severe disease individually when enrolled [according to the guidelines of Government of India, Ministry of Health and Family Welfare”

Comment 2. The ChAdOx1 nCoV-19/Covishield has been withdrawn based on concerns of thrombotic thrombocytopenia. Thus, a relevant question is, “Why are the authors analyzing immune responses from a vaccine withdrawn from the market?”

Reply: India commenced COVID-19 vaccine administration on January 16, 2021. As of March 4, 2023, over 2.2 billion doses of vaccines, including first, second, and precautionary (booster) doses, have been administered across the country. In India, 95% of the eligible population aged 12 and above have received at least one dose, and 88% are fully vaccinated. Covishield constitutes the majority of the approved vaccines administered in India, prompting our interest in studying its impact, together with Covaxin on host immune protection.

Comment 3. The authors analyze virus-neutralization responses using a SARS-CoV2 Surrogate Virus Neutralization test kit. In Figure 2, the data is presented as % inhibition. The authors need to define the meaning of 100% inhibition. How does this compare to titers obtained with virus neutralization test titers? 

Reply : In the context of the SARS-CoV-2 Surrogate Virus Neutralization Test (sVNT), "inhibition" refers to the ability of antibodies present in a sample to block the interaction between the receptor-binding domain (RBD) of the spike protein of SARS-CoV-2 and its receptor on the host cell. When antibodies are present in a sample, they can prevent the RBD on the surrogate virus from binding to its receptor. The degree to which this binding is blocked or inhibited by the antibodies is measured in the assay. Changes are made in lines : 146-150

To answer the other question Both the surrogate virus neutralization test  (sVNT) and virus neutralization test  (VNT) are valuable tools for assessing neutralizing antibody responses against SARS-CoV-2. The sVNT offers speed and scalability but may not directly correlate with protection as definitively as VNT titers, which directly measure virus-neutralizing capacity.

sVNT: This test uses a surrogate virus particle that contains the receptor-binding domain (RBD) of the SARS-CoV-2 spike protein. It measures the ability of antibodies in a sample to inhibit the interaction between this RBD and its receptor, mimicking the virus's entry process.

VNT: The traditional VNT involves using live SARS-CoV-2 virus in a laboratory setting. It tests whether antibodies in a sample can neutralize the actual virus by preventing it from infecting cells in culture.

Comment 4. In Table 1, the values stated for ‘S.No” rows 15-20 are what kind of units? Please define in the Materials and methods.

Reply : As suggested we have added units  in the manuscript. Changes made in the S.No 15-20 table 1

Comment 5. In Figures 3-5, the authors quantify the levels of different subsets of B cells (Fig.3), T cells (Fig. 4) and monocytes (Fig. 5). On the y-axis is the frequency of these cells. For example, the scale may be from 1 to 100. What is missing the frequency of what? Is it 10 or 100 cells per 1,000,000? Without this information, the data has no meaning. This needs to be fixed. Additionally, the experiments in these figures are missing important controls of non-vaccinated, non-infected individuals and individuals vaccinated with an unrelated virus. Finally, while the authors can find some differences, these differences may be due to natural variations in the human population. Functional anti-SARS-CoV-2 assays in various cell populations would be helpful to correlate with these differences. 

Reply : We thank reviewer for this valuable comments we added this information ‘The gating was set by forward and side scatter, and 100,000 gated events were acquired’ in the materials and methods. Changes are made in lines : 119-123

We have now modified as “Eight-color flow cytometry was performed on a FACS Canto II flow cytometer with FACS DIVA software, version 6 (Becton Dickinson). The gating was set by forward and side scatter, and 100,000 gated events were acquired. Data were collected and analyzed using FLOW JO software 10.8.0 (TreeStar, Ashland, OR). Leukocytes were gated using CD45 expression versus side scatter”

Adding a control is an valuable point however in India, where COVID-19 has been widespread, While finding pure controls (individuals completely unexposed and unvaccinated) may be challenging in a pandemic situation. However With reference to the  non-vaccinated and non-infected controls, We have historic controls for that population, which has been studied in our earlier studies several times. For comparison purpose, the mean value of those significantly different analytes has been checked with those historic controls, as recommended by the reviewer. We could see dimnished memory T cells and B cell responses as well as the inflammatory responses among  them, when compared to the study groups. However, We felt that that inference do not have any potential impact towards understanding the host immune response against COVID vaccination. Hence, we have not added to the manuscript for discussion. In addition, the main objective of this study is to study whether vaccinated individuals are more protected than the naturally infected individuals.

Comment 6. In Figure 6, the authors show cytokine levels in vaccinated versus natural infections. The authors show elevated levels of most cytokines in the COVID-19 group. While these authors have a p-value, most levels are minimal at best and are likely not biologically significant. Similar to Figures 3-5, the experiments reported in Figure 6 are missing important controls of non-vaccinated, non-infected individuals. 

Reply: Thanks for bringing this important point most of the measured cytokines which are play an inflammatory role in mediating pathogenesis or host protection, our study population are normal healthy volunteers vaccinated are previously COVID-19 infected hence the inflammatory milieu is predominantly lower in this cohort.  

As we mention in our previous point Adding a control is an valuable point however in India, where COVID-19 has been widespread, While finding pure controls (individuals completely unexposed and unvaccinated) may be challenging in a pandemic situation, finding individuals who have not been exposed to the virus can be challenging due to the high prevalence of infection. We have addressed this point in our previous response.

Minor comments:

  1. Lines 30-32: “COVID-19 (Corona Virus Disease-2019) caused by infection of SARS-CoV-2 (Severe Acute Respiratory Syndrome causing Corona Virus – 2) has been the key reason behind 31 the utmost health crisis all around the world [1].” 

should be changed to,

“COVID-19 (Corona Virus Disease-2019) caused by infection of SARS-CoV-2 (Severe Acute Respiratory Syndrome Coronavirus 2) has been the key reason behind 31 the utmost health crisis all around the world [1].”

Reply : As suggested we have now made changes in the manuscript. Changes are made in lines : 56-58

  1. Lines 41-43: The sentence, “During the natural SARS-CoV-2 infection, immunity which develops results in prompt and efficient immune response, thereby protecting the host.”

should be rewritten as,

During the natural SARS-CoV-2 infection, prompt and efficient innate and adaptive immune responses develop against the virus to protect the host.”

Reply : As suggested we have now made changes in the manuscript. Changes are made in lines : 66-68

  1. Line 72: “enrolment” is spelled “enrollment.”

Reply : As suggested we have now made changes in the manuscript. Changes are made in lines : 96

  1. Line 74:  As used, “Immunology lab” should not be capitalized.

Reply : As suggested we have now made changes in the manuscript. Changes are made in lines : 98

  1. Line 98: Should “1,00000” be “100,000”?

Reply : As suggested we have now made changes in the manuscript. Changes are made in lines : 121

  1. Lines 108-109: The sentence “N = 44 samples were analyzed per batch in duplicates, and lab personal were blinded to the clinical groups.” 

should be re-written as, 

Forty-four samples were analyzed in duplicate and lab personnel were blinded to clinical groups.

Reply : As suggested we have now made changes in the manuscript. Changes are made in lines : 131-132

  1. Lines 114-116: The sentences, “The cut-off value for SARS-CoV-2 IgG, according to the manufacturer, IgG concentrations more than or equal to 10.00 AU/mL was considered as positive and <10.00 115 AU/mL was considered as non-reactive,”  

should be rewritten as,

 “The cut-off value for SARS-CoV-2 IgG (according to the manufacturer) was >10 AU/mL, IgG concentrations >10.00 AU/mL was considered as positive and  <10.00 AU/mL considered as non-reactive.”

Reply : As suggested we have now made changes in the manuscript. Changes are made in lines : 137-139

  1. Lines 130-134: The sentence, “COVID-19 vaccinated individuals: Individuals after Completion of two doses of Covaxin or Covishield vaccine with no previous RT-PCR or Rapid antigen test confirmation, for last six months and COVID-19 naturally infected individualsIndividuals with post PCR confirmation within 30 days, who had either asymptomatic, mild and moderate but not, severe disease individually are enrolled…”

should be rewritten as,

 “COVID-19 vaccinated individuals were those who completed two doses of Covaxin or Covishield vaccine with no previous RT-PCR or Rapid antigen test confirmation, for the last six months and COVID-19 naturally infected individuals were those with a post-PCR confirmation within 30 days, who were asymptomatic or had mild and moderate, but not, severe disease individually when enrolled…..”

Reply : As suggested we have now made changes in the manuscript. Changes are made in lines : 162-169

The authors discuss “COVID naturally infected” individuals as having asymptomatic, mild, or moderate infections of SARS-CoV-2 yet in Figure 1, the flowchart only mentions individuals having mild infections. Is this correct? If so, this should be stated in the Materials and Methods section. Why didn’t the authors analyze individuals with asymptomatic and moderate infections? These also would be important to examine.

Reply : As suggested we have now included this information in the materials and methods section. Changes are made in lines : 94-95

  1. Lines 175,  211, 242: “p value….” should be changed to “Thep-value……”

Lines 251-253:  The sentence, “The immunological pathogenicity of COVID-19 is difficult and may be correlated with the virulence of SARS-CoV-2 and the lack of the temporal coordination between innate and adaptive immune responses [17].” 

would sound better as,

The immunopathogenesis of COVID-19 is complex and may be associated with the virulence of SARS-CoV-2 and the lack of temporal coordination between the innate and adaptive immune responses [17]. 

Reply : As suggested we have now made changes in the manuscript. Changes are made in lines : 257-259

  1. Line 255:  “Covid-19” should be “COVID-19.”

Reply : As suggested we have now made changes in the manuscript. Changes are made in lines : 259

Reviewer 2 Report

Comments and Suggestions for Authors

Elucidating the Immune Response to SARS-CoV2:Natural Infection versus Vaccination in a South Indian Population

In this manuscript, Agalya et al. determine the immune differences (cytokines, antibodies, B/T cells, memory) between individuals that were vaccinated against SARS-CoV2 and non-vaccinated individuals that had SARS-CoV2 (mild/moderate symptoms). Overall, this work highlights key differences in vaccinated vs. naturally infected individuals and is very important work.

Specific comments:

Figure 6, IL-5 panel-is this mislabeled as significant? The means/dots look the same between the 2 groups. While I am not an expert in statistics and it is hard to determine if metrics of a t-test are not met, if this is significant when there is no difference if this is the correct statistical test to use.

While some future studies are included in lines 367-371, what about determining if there was infection in these 2 groups after this analysis (aka-do these differences, especially in memory, mean anything for infection/severity of later infection). This could be posed as a question (especially if following the 2 groups isn’t possible. I think that is key/what the readers will want to know.

Author Response

In this manuscript, Agalya et al. determine the immune differences (cytokines, antibodies, B/T cells, memory) between individuals that were vaccinated against SARS-CoV2 and non-vaccinated individuals that had SARS-CoV2 (mild/moderate symptoms). Overall, this work highlights key differences in vaccinated vs. naturally infected individuals and is very important work.

Specific comments:

Figure 6, IL-5 panel-is this mislabelled as significant? The means/dots look the same between the 2 groups. While I am not an expert in statistics and it is hard to determine if metrics of a t-test are not met, if this is significant when there is no difference if this is the correct statistical test to use.

Reply : We thank reviewer for  brings this import point, as the graph was presented in the log scale the actual differences are nor well displayed, however we do see there is an statical differences are seen between the both the groups. Please find the below IL-5 data shown in the linear scale

                                                                                   IL-5

While some future studies are included in lines 367-371, what about determining if there was infection in these 2 groups after this analysis (aka-do these differences, especially in memory, mean anything for infection/severity of later infection). This could be posed as a question (especially if following the 2 groups isn’t possible. I think that is key/what the readers will want to know.

Reply: We thank reviewer for the valuable comments, as suggested we now added more information in  regard in the manuscript. Changes are made in lines : 392-397

Reviewer 3 Report

Comments and Suggestions for Authors

The SARS-CoV-2 coronavirus emerged in December, 2019 in Wuhan, China and rapidly spread across the world causing the Covid-19 epidemic that took millions of lives, as well as being responsible for devastating economic and social issues that, in some respects, are still ongoing.  However, it is clear that the primary factor in bringing the virus under control were the herculean efforts of the scientific community in the development and distribution of efficacious Covid-19 vaccines in record time. However, in addition to the extensive mortality, the virus caused a range of disease severity in patients who somehow survived without vaccination. 

This study compares the immune responses in the South Indian population resulting from natural infection by the virus and immunization by two vaccines that were widely administered in the country, Covaxin, a whole virion inactivated vaccine formulated with a toll-like receptor ligand adsorbed to alum and Covishield, a recombinant, replication-deficient chimpanzee adenovirus vector encoding the viral spike glycoprotein.  The efficacy of both vaccines was extremely high, 80% from the former and nearly 90% from the latter.  The group conducts a highly comprehensive evaluation of the various components of the immune response, comparing each in vaccinated and naturally infected individuals.  To their credit, they evaluate the response to, not only the ancestral strain of the virus, but also multiple subsequently emerging variants, including the Delta, Omicron, Beta and Alpha lineages.

The data presented convincingly demonstrate that the vaccines elicit a more effective and longer lasting classical memory B cell response, as well as that of activated memory B cells and plasma cells in comparison to the same parameters measured in infected patients.  The vaccine vs. infection comparison is less clear with respect to the cellular arm of the response.  Although vaccinated individuals exhibited increased numbers of classical and intermediate monocytes, the non-classical components of this response were significantly decreased in the vaccinated population.  Similarly, the infected group also exhibited higher levels of proinflammatory cytokines than the vaccinated group.  Overall, the study reinforces the importance and efficacy of Covid-19 vaccination, as the vaccine-induced immune memory response elicits a longer lasting and more protective response than the natural infection.

This is considered a strong, well-designed and appropriately interpreted study that contributes to our understanding of the immune response in infected and vaccinated individuals.  Most importantly, the study confirms the benefits of Covid-19 vaccination, which can be highly impactful in the present climate of vaccine weariness in the population.  One caveat that should be mentioned and perhaps acceded to by the authors is the relatively small number of participants in the study population with only 53 vaccinated and 26 unvaccinated individuals.  The authors should mention this limitation at some point in the manuscript.

Author Response

Reviewer 3

The SARS-CoV-2 coronavirus emerged in December, 2019 in Wuhan, China and rapidly spread across the world causing the Covid-19 epidemic that took millions of lives, as well as being responsible for devastating economic and social issues that, in some respects, are still ongoing.  However, it is clear that the primary factor in bringing the virus under control were the herculean efforts of the scientific community in the development and distribution of efficacious Covid-19 vaccines in record time. However, in addition to the extensive mortality, the virus caused a range of disease severity in patients who somehow survived without vaccination. 

This study compares the immune responses in the South Indian population resulting from natural infection by the virus and immunization by two vaccines that were widely administered in the country, Covaxin, a whole virion inactivated vaccine formulated with a toll-like receptor ligand adsorbed to alum and Covishield, a recombinant, replication-deficient chimpanzee adenovirus vector encoding the viral spike glycoprotein.  The efficacy of both vaccines was extremely high, 80% from the former and nearly 90% from the latter.  The group conducts a highly comprehensive evaluation of the various components of the immune response, comparing each in vaccinated and naturally infected individuals.  To their credit, they evaluate the response to, not only the ancestral strain of the virus, but also multiple subsequently emerging variants, including the Delta, Omicron, Beta and Alpha lineages.

The data presented convincingly demonstrate that the vaccines elicit a more effective and longer lasting classical memory B cell response, as well as that of activated memory B cells and plasma cells in comparison to the same parameters measured in infected patients.  The vaccine vs. infection comparison is less clear with respect to the cellular arm of the response.  Although vaccinated individuals exhibited increased numbers of classical and intermediate monocytes, the non-classical components of this response were significantly decreased in the vaccinated population.  Similarly, the infected group also exhibited higher levels of proinflammatory cytokines than the vaccinated group.  Overall, the study reinforces the importance and efficacy of Covid-19 vaccination, as the vaccine-induced immune memory response elicits a longer lasting and more protective response than the natural infection.

This is considered a strong, well-designed and appropriately interpreted study that contributes to our understanding of the immune response in infected and vaccinated individuals.  Most importantly, the study confirms the benefits of Covid-19 vaccination, which can be highly impactful in the present climate of vaccine weariness in the population.  One caveat that should be mentioned and perhaps acceded to by the authors is the relatively small number of participants in the study population with only 53 vaccinated and 26 unvaccinated individuals.  The authors should mention this limitation at some point in the manuscript.

Reply : We thank  a reviewer for their valuable comments and feedback you provided. Your comments were incredibly valuable in guiding revisions and improving the overall quality of the manuscript. As suggested we added the study limitation in the manuscript. Changes are made in lines : 382-284

Reviewer 4 Report

Comments and Suggestions for Authors

The authors performed a comprehensive study on the immune response to SARS-CoV2 in a South Indian population. They focused on the comparison between the immunity conferred by vaccines and naturally acquired immunity, including antibody levels, cellular parameters, and cytokine response. Their research is valuable for understanding the mechanism of the protection of the COVID19 vaccines.

Here are only one suggestion concerning their study:

1. The authors should provide more details on the method to test the neutralizing ability of plasma antibody (Line 116-120). did they purify the antibody? How can the authors excluded the neutralizing background of plasma?

Author Response

Reviewer 4:

The authors performed a comprehensive study on the immune response to SARS-CoV2 in a South Indian population. They focused on the comparison between the immunity conferred by vaccines and naturally acquired immunity, including antibody levels, cellular parameters, and cytokine response. Their research is valuable for understanding the mechanism of the protection of the COVID19 vaccines.

Here are only one suggestion concerning their study:

The authors should provide more details on the method to test the neutralizing ability of plasma antibody (Line 116-120). did they purify the antibody? How can the authors excluded the neutralizing background of plasma?

Reply : We thank reviewer for the comments as suggested we have now added more information about the Virus neutralisation assay. Changes are made in lines : 139-150

Reviewer 5 Report

Comments and Suggestions for Authors

The work entitled “Elucidating the Immune Response to SARS-CoV2: Natural Infection versus Vaccination in a South Indian Population” provides an analysis of immune parameters elicited by natural infection vs two vaccines in a prospective cross sectional study conducted in South Indian population. 

 The paper is well written, clearly and understandable also by a non professional audience. Nevertheless, it needs a different kind of analysis, among other minor/major changes to be accepted for publication.

Major:

1) The two types of vaccines analyzed together, should be analyzed separately also, since these are not the same kind of vaccine (live attenuated / adyuvantated vs unrelated viral engineered vector delivered).

Minor:

2) The names of the vaccines analyzed should appear in the title of the manuscript to make clear what was specifically analyzed.

3) Please reformulate the sentence from lines 18 to 20.

4)Lines 70 to 72: How you justify that down selecting asymptomatic to moderate COVID infections you are not biased the study and somehow the differences in the results?

5)Table 1 should discriminate the results between vaccines types.

6) In line 98 the number you entered is 100.000 or 1.000.000? Please clarify.

7) Figure 1 should discriminate the results between vaccines types.

8) In line 196 Figure is a different figure number (not number 3) please correct.

9) In lines 196 to 199  please justify the markers used for the different cell subsets.

10) In lines 208 and 213 please choose a different word (e.g: enhanced, decreased or increased).

11) Figure 6: Please plot these in a spider plot, again, discriminate the two vaccines involved.

12) Incorporate to the discussion the type of vaccine involved and the type of immune response attained vs other vaccine platforms that use mRNA and its advantages / disadvantages.

13) In references, please include the citation about "SARS-CoV-2 and Immunity: Natural Infection Compared

with Vaccination" Vespa, S et al. Int. J. Mol. Sci. 2022, 23, and discuss it

To published the paper, all the corrections and suggestions should be implemented.

My best regards,

The reviewer.

Author Response

Reviewer 5

The work entitled “Elucidating the Immune Response to SARS-CoV2: Natural Infection versus Vaccination in a South Indian Population” provides an analysis of immune parameters elicited by natural infection vs two vaccines in a prospective cross sectional study conducted in South Indian population. 

 The paper is well written, clearly and understandable also by a non-professional audience. Nevertheless, it needs a different kind of analysis, among other minor/major changes to be accepted for publication.

Major:

1) The two types of vaccines analyzed together, should be analyzed separately also, since these are not the same kind of vaccine (live attenuated / adyuvantated vs unrelated viral engineered vector delivered).

Reply:  We agree with the reviewer we do analysed separately by vaccine type, however we don’t see any differences between both the vaccines hence we combined and presented as a single group.   

Minor:

2) The names of the vaccines analyzed should appear in the title of the manuscript to make clear what was specifically analyzed.

 Reply : As suggested we now modified in the manuscript. Changes are made in lines : 2

As suggested we now modified as “Elucidating the Immune response to SARS-CoV2: Natural infection Versus Covaxin/Covishield Vaccination in a South Indian population”

3) Please reformulate the sentence from lines 18 to 20.

Reply : As suggested we have now reformulate the sentence. Changes are made in lines : 37-41

As suggested we now modified as “COVID-19 vaccination stimulates heightened levels of IgG antibodies against the original strain of SARS-CoV-2, as well as increased binding to the spike protein and neutralizing antibody levels. This enhanced response extends to variant lineages such as B.1.617.2 (Delta, India), B.1.1.529 (Omicron, India), B.1.351 (Beta, South Africa), and B.1.1.7 (Alpha, UK)”

4)Lines 70 to 72: How you justify that down selecting asymptomatic to moderate COVID infections you are not biased the study and somehow the differences in the results?

Reply : We have done sub-analysis between asymptomatic to moderate COVID infections we don’t see any differences among them.

5)Table 1 should discriminate the results between vaccines types.

Reply : We have now added this table as an supplementary file

6) In line 98 the number you entered is 100.000 or 1.000.000? Please clarify.

 Reply: We have now modified accordingly. Changes are made in lines : 121

7) Figure 1 should discriminate the results between vaccines types.

Reply : As previously noted in response to earlier comments, since no discernible difference was observed between the two types of vaccines, we have chosen to maintain them as a combined group.

8) In line 196 Figure is a different figure number (not number 3) please correct.

 Reply : We apologies for the typo error, we have now modified accordingly

9) In lines 196 to 199  please justify the markers used for the different cell subsets.

 Reply : As suggested we have now modified accordingly in the manuscript. Changes are made in lines : 213-217

As suggested we now modified as “To assess the ex-vivo phenotype of CD4+ memory T cell subsets (CD45RA, CCR7, CD95, and CD28 are markers found on T cells, CD45RA: Marks naive T cells that haven't encountered specific threats yet, CCR7: Guides T cells to lymph nodes where they can respond to threats, CD95 : Controls T cell survival and prevents overactive immune responses and CD28: Helps activate T cells by interacting with other immune cells) following COVID-19 vaccination and natural infection,”

10) In lines 208 and 213 please choose a different word (e.g: enhanced, decreased or increased).

 Reply : Since we observed an increase in one cell type and a decrease in another, we used the term "altered" instead of specific terms.

11) Figure 6: Please plot these in a spider plot, again, discriminate the two vaccines involved.

Reply : As previously noted in response to earlier comments, since no discernible difference was observed between the two types of vaccines, we have chosen to maintain them as a combined group.

12) Incorporate to the discussion the type of vaccine involved and the type of immune response attained vs other vaccine platforms that use mRNA and its advantages / disadvantages.

Reply : As suggested we have now made these changes in the manuscript. Changes are made in lines : 261-274

As suggested we now modified as “Inactivated vaccines are considered safe because they do not contain live virus, eliminating the risk of causing disease in vaccinated individuals and also offer safety, stability, and suitability for vulnerable populations but may require multiple doses and adjuvants for optimal efficacy. These inactivated vaccines stimulate immune responses primarily through the production of antibodies by B cells. These responses are crucial for providing protection against specific pathogens without causing the disease itself [18, 19]. Recombinant adenovirus vector vaccines can induce potent immune responses because adenoviruses are highly immunogenic. They can stimulate both cellular (T cell-mediated) and humoral (antibody-mediated) immune responses, providing comprehensive protection against pathogens. Adenovirus vectors can be engineered to carry genes encoding specific antigens from various pathogens, however any people have pre-existing immunity to adenoviruses due to prior natural infections or vaccinations. This immunity can potentially reduce the effectiveness of adenovirus vector vaccines by neutralizing the vector before it can deliver the vaccine antigen [20, 21].”

13) In references, please include the citation about "SARS-CoV-2 and Immunity: Natural Infection Comparedwith Vaccination" Vespa, S et al. Int. J. Mol. Sci. 2022, 23, and discuss it

 Reply : As suggested we have cited and discussed this article in our manuscript. Changes are made in lines : 280-283

As suggested we have now modified as “A recent study also reported that Overall, while natural infection can provide immunity against SARS-CoV-2, COVID-19 vaccination offers a more reliable and controlled means of achieving robust and durable protection against the virus and its variants [22]”

Round 2

Reviewer 1 Report

Comments and Suggestions for Authors

Comment 1. This has been addressed.

Comment 2. The ChAdOx1 nCoV-19/Covishield has been withdrawn based on concerns of thrombotic

thrombocytopenia. Thus, a relevant question is, “Why are the authors analyzing immune responses

from a vaccine withdrawn from the market?”

Reply: India commenced COVID-19 vaccine administration on January 16, 2021. As of March 4, 2023,

over 2.2 billion doses of vaccines, including first, second, and precautionary (booster) doses, have been

administered across the country. In India, 95% of the eligible population aged 12 and above have

received at least one dose, and 88% are fully vaccinated. Covishield constitutes the majority of the

approved vaccines administered in India, promoting our interest in studying its impact, together with

Covaxin on host immune protection.

Reviewer response: The authors need to acknowledge this in the manuscript. However, studying the immune responses to a vaccine that no longer is on the market will likely not contribute field.

Comment 3. The authors analyze virus-neutralization responses using a SARS-CoV2 Surrogate Virus

Neutralization test kit. In Figure 2, the data is presented as % inhibition. The authors need to define

the meaning of 100% inhibition. How does this compare to titers obtained with virus neutralization

test titers?

Reply : In the context of the SARS-CoV-2 Surrogate Virus Neutralization Test (sVNT), "inhibition" refers

to the ability of antibodies present in a sample to block the interaction between the receptor-binding

domain (RBD) of the spike protein of SARS-CoV-2 and its receptor on the host cell. When antibodies

are present in a sample, they can prevent the RBD on the surrogate virus from binding to its receptor.

The degree to which this binding is blocked or inhibited by the antibodies is measured in the assay.

Changes are made in lines : 146-150

Reviewer response: The concern here was that the authors talk about % inhibition. You can’t discuss % inhibition without having a 100% inhibition control and a 0% inhibition control. This still needs to be clarified.

Comment 4. This has been addressed.

Comment 6. In Figure 6, the authors show cytokine levels in vaccinated versus natural infections. The

authors show elevated levels of most cytokines in the COVID-19 group. While these authors have a p value, most levels are minimal at best and are likely not biologically significant. Similar to Figures 3-5,

the experiments reported in Figure 6 are missing important controls of non-vaccinated, non-infected

individuals.

Reply: Thanks for bringing this important point most of the measured cytokines which are play an

inflammatory role in mediating pathogenesis or host protection, our study population are normal

healthy volunteers vaccinated are previously COVID-19 infected hence the inflammatory milieu is

predominantly lower in this cohort.

As we mention in our previous point Adding a control is an valuable point however in India, where

COVID-19 has been widespread, While finding pure controls (individuals completely unexposed and

unvaccinated) may be challenging in a pandemic situation, finding individuals who have not been

exposed to the virus can be challenging due to the high prevalence of infection. We have addressed

this point in our previous response.

Reviewer response: While I appreciate the author's problems for a control population (unexposed, unvaccinated), I don’t believe we should support science without the necessary controls as it will likely  lead to equivocal findings. As far as this reviewer is concerned, these controls do not necessarily have to be from India but rather could come from other countries. What I’m saying is an uninfected, unvaccinated humans could come from anywhere in the world. This also pertains to Comment 5.

Minor comments: These have all been corrected.

Comments on the Quality of English Language

see comments above

Author Response

Comment 1. This has been addressed.

Comment 2. The ChAdOx1 nCoV-19/Covishield has been withdrawn based on concerns of thrombotic thrombocytopenia. Thus, a relevant question is, “Why are the authors analyzing immune responses from a vaccine withdrawn from the market?”

Reply: India commenced COVID-19 vaccine administration on January 16, 2021. As of March 4, 2023,over 2.2 billion doses of vaccines, including first, second, and precautionary (booster) doses, have been administered across the country. In India, 95% of the eligible population aged 12 and above have received at least one dose, and 88% are fully vaccinated. Covishield constitutes the majority of the approved vaccines administered in India, promoting our interest in studying its impact, together with Covaxin on host immune protection.

Reviewer response: The authors need to acknowledge this in the manuscript. However, studying the immune responses to a vaccine that no longer is on the market will likely not contribute field. 

Reply (Rev 2) : As suggested by the reviewer we have now added this information in the manuscript as suggested we now added this information in the manuscript as ” India commenced COVID-19 vaccine administration on January 16, 2021. As of March 4, 2023,over 2.2 billion doses of vaccines, including first, second, and precautionary (booster) doses, have been administered across the country. In India, 95% of the eligible population aged 12 and above have received at least one dose, and 88% are fully vaccinated. Covishield constitutes the majority of the approved vaccines administered in India, promoting our interest in studying its impact, together with Covaxin on host immune protection” Changes made in the lines 60-66

Comment 3. The authors analyze virus-neutralization responses using a SARS-CoV2 Surrogate Virus Neutralization test kit. In Figure 2, the data is presented as % inhibition. The authors need to define the meaning of 100% inhibition. How does this compare to titers obtained with virus neutralization test titers?

Reply : In the context of the SARS-CoV-2 Surrogate Virus Neutralization Test (sVNT), "inhibition" refers to the ability of antibodies present in a sample to block the interaction between the receptor-binding domain (RBD) of the spike protein of SARS-CoV-2 and its receptor on the host cell. When antibodies are present in a sample, they can prevent the RBD on the surrogate virus from binding to its receptor. The degree to which this binding is blocked or inhibited by the antibodies is measured in the assay. Changes are made in lines : 146-150

Reviewer response: The concern here was that the authors talk about % inhibition. You can’t discuss % inhibition without having a 100% inhibition control and a 0% inhibition control. This still needs to be clarified.

Reply (Rev 2): To ensure the validity of our results, we conduct each assay with positive and negative controls provided in the kit. The net optical density (OD450) for both controls must fall within specified ranges, as detailed in the table below. If the OD450 values of these controls do not meet the criteria outlined in the table, the test is deemed invalid and must be repeated. This strict adherence to control thresholds ensures the reliability and accuracy of our sVNT assay results.

Items

OD450 value

Control Result for Valid Assay

Quality Control

>1.0

Negative Control

<0.3

Positive Control

Interpretation of Results

The positive cutoff and negative cutoff for SARS-CoV-2 neutralizing antibody detection  are used for interpretation of the inhibition rate. We determine the result of the sample by comparing the inhibition rate to the following table.

Items

Cutoff

Result

Interpretation

SARS-CoV-2 neutralizing antibody

>20%

Positive

SARS-CoV-2 neutralizing antibody detected

<20%

Negative

No detectable SARS-CoV-2 neutralizing antibody

Comment 4. This has been addressed.

Comment 6. In Figure 6, the authors show cytokine levels in vaccinated versus natural infections. The authors show elevated levels of most cytokines in the COVID-19 group. While these authors have a p value, most levels are minimal at best and are likely not biologically significant. Similar to Figures 3-5, the experiments reported in Figure 6 are missing important controls of non-vaccinated, non-infected individuals.

Reply: Thanks for bringing this important point most of the measured cytokines which are play an inflammatory role in mediating pathogenesis or host protection, our study population are normal healthy volunteers vaccinated are previously COVID-19 infected hence the inflammatory milieu is predominantly lower in this cohort.

As we mention in our previous point Adding a control is an valuable point however in India, where COVID-19 has been widespread, While finding pure controls (individuals completely unexposed and unvaccinated) may be challenging in a pandemic situation, finding individuals who have not been exposed to the virus can be challenging due to the high prevalence of infection. We have addressed this point in our previous response.

Reviewer response: While I appreciate the author's problems for a control population (unexposed, unvaccinated), I don’t believe we should support science without the necessary controls as it will likely  lead to equivocal findings. As far as this reviewer is concerned, these controls do not necessarily have to be from India but rather could come from other countries. What I’m saying is an uninfected, unvaccinated humans could come from anywhere in the world. This also pertains to Comment 5.

Reply (Rev 2) : As previously mentioned, obtaining pure controls (individuals completely unexposed and unvaccinated) during a pandemic is challenging. Additionally, acquiring controls from other countries is not feasible under current circumstances. Therefore, we utilized historic controls (pre-pandemic healthy individuals) to compare between vaccinated individuals and those who have had COVID-19.

A separate data was analysed in compassion to healthy controls and information are added in the Sup. Table 2. Changes are made in the manuscript in the lines of 182-185 and 258-262

Minor comments: These have all been corrected

Reviewer 5 Report

Comments and Suggestions for Authors

Dear Authors:

Second round of revision for the work: “Elucidating the Immune Response to SARS-CoV2: Natural Infection versus Vaccination in a South Indian Population”.

Response to Author’s comments (R=reviewer; A=Authors):

1)

R: 4) Lines 70 to 72: How you justify that down selecting asymptomatic to moderate COVID infections you are not biased the study and somehow the differences in the results?

A: Reply: We have done sub-analysis between asymptomatic to moderate COVID infections we don’t see any differences among them. 

R: Please show the mentioned analysis at least in supplementary materials. 

2)

R: 5) Table 1 should discriminate the results between vaccines types.

A: Reply : We have now added this table as an supplementary file

R: Please move it to the main body of the text, replace the current table 1 with it.

3)

R: 7) Figure 1 should discriminate the results between vaccines types.

A: Reply : As previously noted in response to earlier comments, since no discernible difference was observed between the two types of vaccines, we have chosen to maintain them as a combined group.

R: Please remake the figure with the requested changes (alternatively you can add a figure with the two vaccines as different groups or treatments in each panel, in supplementary materials). 

You should do this for all the figures showing cell populations (Figure 2, 3, 4, and 5) and for figure 6 also with cytokine levels (in spider plot as suggested if possible). This is because you showed now in Supp.Table 1 that the WBC populations rendered in circulation after vaccination differ between vaccines. 

4)

R: 9) In lines 196 to 199 please justify the markers used for the different cell subsets.

A: Reply : As suggested we have now modified accordingly in the manuscript. Changes are made in lines : 213-217

As suggested we now modified as “To assess the ex-vivo phenotype of CD4+ memory T cell subsets (CD45RA, CCR7, CD95, and CD28 are markers found on T cells, CD45RA: Marks naive T cells that haven't encountered specific threats yet, CCR7: Guides T cells to lymph nodes where they can respond to threats, CD95 : Controls T cell survival and prevents overactive immune responses and CD28: Helps activate T cells by interacting with other immune cells) following COVID-19 vaccination and natural infection,”

R: What you wrote is basic Immunological knowledge: …” CD45RA, CCR7, CD95, and CD28 are markers found on T cells”...  What I asked is why you used this markers for T cell repertoires, - for instance, instead of classical CD44 & CD62L-? Please craft an explanation targeted to the reader of Viruses. Thanks. 

Upon modification of figures and changes in explanations the paper should be ready for publication. My best.

The reviewer.

Author Response

Dear Authors:

Second round of revision for the work: “Elucidating the Immune Response to SARS-CoV2: Natural Infection versus Vaccination in a South Indian Population”

Response to Author’s comments (R=reviewer; A=Authors):

 R: 4) Lines 70 to 72: How you justify that down selecting asymptomatic to moderate COVID infections you are not biased the study and somehow the differences in the results?

 A: Reply: We have done sub-analysis between asymptomatic to moderate COVID infections we don’t see any differences among them. 

 R: Please show the mentioned analysis at least in supplementary materials. 

Reply Rev 2 : As suggested by the reviewer we have now added this information in the supplementary table 3

R: 5) Table 1 should discriminate the results between vaccines types.

A: Reply : We have now added this table as an supplementary file

R: Please move it to the main body of the text, replace the current table 1 with it.

Reply Rev2 : Since the primary focus of the manuscript is to compare immune responses between vaccinated individuals and those with COVID-19, we will include this data in the supplementary table, alongside other datasets comparing different vaccine types.

R: 7) Figure 1 should discriminate the results between vaccines types.

A: Reply : As previously noted in response to earlier comments, since no discernible difference was observed between the two types of vaccines, we have chosen to maintain them as a combined group.

R: Please remake the figure with the requested changes (alternatively you can add a figure with the two vaccines as different groups or treatments in each panel, in supplementary materials). 

You should do this for all the figures showing cell populations (Figure 2, 3, 4, and 5) and for figure 6 also with cytokine levels (in spider plot as suggested if possible). This is because you showed now in Supp.Table 1 that the WBC populations rendered in circulation after vaccination differ between vaccines. 

Reply Rev 2: We thank reviewer for the comment as suggested we have made separate figures of the vaccine types for Figures 2, 3, 4, 5 and 6, Now all these information are added in the Sup. Figures 2-6

R: 9) In lines 196 to 199 please justify the markers used for the different cell subsets.

A: Reply : As suggested we have now modified accordingly in the manuscript. Changes are made in lines : 213-217

As suggested we now modified as “To assess the ex-vivo phenotype of CD4+ memory T cell subsets (CD45RA, CCR7, CD95, and CD28 are markers found on T cells, CD45RA: Marks naive T cells that haven't encountered specific threats yet, CCR7: Guides T cells to lymph nodes where they can respond to threats, CD95 : Controls T cell survival and prevents overactive immune responses and CD28: Helps activate T cells by interacting with other immune cells) following COVID-19 vaccination and natural infection,”

R: What you wrote is basic Immunological knowledge: …” CD45RA, CCR7, CD95, and CD28 are markers found on T cells”...  What I asked is why you used this markers for T cell repertoires, - for instance, instead of classical CD44 & CD62L-? Please craft an explanation targeted to the reader of Viruses. Thanks. 

Reply Rev2 : We thank the reviewer for the valuable comment, as suggested we now modifed as “In our current study, we focused on using CD45RA and CCR7 to primarily delineate memory T cell differentiation states, rather than relying on CD44 and CD62L which predominantly indicate T cell activation and tissue homing capabilities. The rationale behind this choice lies in the critical role of CD45RA and CCR7 as markers for defining distinct subsets of memory T cells. These markers provide essential insights into the differentiation, functional capabilities, and immune response roles of memory T cells. Furthermore, their utilization aids in deepening our fundamental understanding of adaptive immune responses in both health and disease contexts.” Changes made in the lines 341-348

Upon modification of figures and changes in explanations the paper should be ready for publication. My best.

The reviewer.
